# Representation Learning for Financial Time-Series Forecasting

## Abstract

The accurate forecasting of financial time series remains a significant challenge due to the stochastic nature of the underlying data. To improve prediction accuracy, feature engineering has become a vital aspect of forecasting financial assets. However, engineering features manually often requires domain expertise. We propose to utilise an automated feature generation architecture, Contrastive Predictive Coding (CPC), to generate embeddings as input to improve the performance of downstream financial time series forecasting models. To benchmark the effectiveness of our approach, we evaluate forecasting models on predicting the next day's log return on various foreign exchange markets with and without embeddings. Finally, we assess our CPC architecture by employing the same trained encoder on different currency pairs and calculating the Sharpe ratio of our strategies.

## 1 Introduction

Analysing financial time series is an integral component of stock trading, risk management and econometrics. Auto-regressive integrated moving averages and linear regression are traditionally used and serve as the foundation for many modern forecasting techniques. However, these methods often falter when confronted with the highly volatile and stochastic nature of financial datasets, where the assumptions of stationarity and linearity do not hold (1).

Advancements in machine learning have allowed for more complex temporal patterns to be captured for both analysis and forecasting of time series (2). Deep Learning (DL) and neural networks enable the extraction of complex patterns and representations which have the potential to enhance predictive accuracy. Notably, Long Short-Term Memory (LSTM) networks have been shown to be able to forecast time series accurately to a degree (3).

LSTMs still struggle when attempting to forecast non-deterministic time series. This is partially due to the large amount of data required for the LSTM to learn more complex patterns. Feature engineering is typically utilised to supplement models with further data but this is a time-consuming step that requires extensive domain knowledge and can introduce bias.

We propose to implement Contrastive Predictive Coding (CPC) architecture as presented by Aaron van den Oord et al. to extract features from financial time series (4). To assess the performance of this approach, linear regression and LSTM models are trained with and without the CPC-generated embeddings and compared based on accuracy metrics. Several baselines models such as the persistence model and naive mean model are implemented as a baseline. By conducting this comparative analysis, we aim to evaluate the relative performance and potential advantages of CPC in high-quality embeddings based unpredictable and highly variable stochastic financial time series.

### 1.1 Literature Overview

(4) introduces CPC as an innovative approach for unsupervised learning. CPC leverages a self-supervised learning framework that predicts future observations in a latent space while contrasting correct futures against negative samples. This approach enables the extraction of meaningful patterns from large datasets without the need for extensive labelled data. CPC's strength lies in its ability to learn compact and informative representations, making it particularly useful for applications where labelled data is scarce. However, CPC was experimented on a deterministic environment and has yet to be applied successfully to financial time series that exhibit stochastic behaviour.

(5) proposes a novel deep learning framework, LSTNet, for multivariate time series forecasting. LSTNet attempts to capture both short-term local dependencies and long-term patterns in time series data utilising convolutional neural networks and recurrent neural networks in combination. In addition, a recurrent-skip mechanism along with an autoregressive component is implemented for the handling of long-term dependencies and scaling issues. Despite the model outperforming benchmarks on certain non-stochastic datasets, it failed to statistically outperform baselines on the one financial dataset it was tested on.

(6) introduces a novel approach called Time-series Representational Learning through CPC (TRL-CPC) for detecting anomalies in multivariate time series data. The proposed method presents an effective way to learn representations using the CPC architecture and a non-linear transformation function. The datasets utilised in this study are not similar to financial time series data, which are notoriously more difficult to predict due to having higher levels of randomness and volatility. Furthermore, the dataset used includes more than one feature, which makes evaluating the CPC architecture's impact on the results more difficult.

The lack of evaluating automated features generated by the CPC architecture in a stochastic environment is what forms the basis of our research.

## 2 METHODS

Here we outline our methodology used to investigate the effectiveness of CPC embeddings in improving financial time series forecasting. Our experiment framework consists of data sourcing and preprocessing, baseline model development, neural network implementation, and the CPC architecture.

### 2.1 PARAMETERS

For our experiment, the following variables were kept constant:

- **windows = 10:** The number of windows in each input to the CPC model.
- **timesteps = 25:** The length of each window.
- **features = 1:** The number of features in the input data (the closing price).
- **code size = 32:** The dimensionality of the encoded representation, also known as the size of the latent space.
- **batch size = 512:** The number of samples processed by the model in one forward/backward pass during training.

### 2.2 DATA SOURCE AND PREPROCESSING

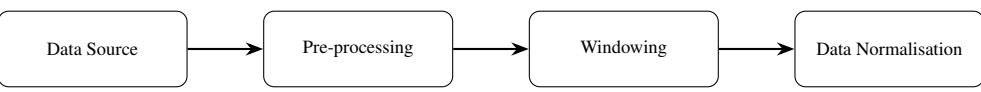

Figure 1: Data Preprocessing Pipeline

#### 2.2.1 DATA SOURCE

The dataset employed in this study and for the training of the CPC architecture comprises daily closing prices of the US Dollar versus the Japanese Yen exchange rate (USDJPY). In addition, our encoder trained on USDJPY is then domain-adapted on two other datasets: the US Dollar versus the Singaporean Dollar (USDSGD) and the Euro versus the Great British Pound (EURGBP). All data was downloaded in CSV format, obtained from Yahoo Finance (7), and subsequently processed to prepare it for analysis. Data from November 1996 till August 2024 was utilised.

### 2.2.2 DATA PREPROCESSING

**Log Returns Calculation:** The raw closing prices were transformed into log returns to stabilise the variance and achieve stationarity. This is a prerequisite for many statistical and machine-learning models due to their assumption of the input data distribution being consistent over time (8). The log return, $r_t$, at time $t$ was calculated as follows:

$$r_t = \log\left(\frac{P_t}{P_{t-1}}\right) \tag{1}$$

where $P_t$ denotes the closing price at time $t$.

**Data Normalisation:** The data was then normalised using min-max scaling. This normalisation is beneficial when training neural networks as it improves convergence during training (due to gradient-based optimisation algorithms) and ensures compatibility with activation functions, to name a few (8). The normalised value, $x_t$, was computed as:

$$x_t = 2\left(\frac{x_t - \min(x)}{\max(x) - \min(x)}\right) - 1 \tag{2}$$

where $\min(x)$ and $\max(x)$ represent the minimum and maximum values of the time series, respectively.

**Sliding Window Approach:** The sliding window technique was utilised to convert the normalised time series data into overlapping sub-sequences, called "windows." Each window has 250 timesteps (nearly one year of trading days) and is slid across the data at a fixed length of 1.

The pseudocode for the sliding window approach is provided below:

```
FUNCTION create_windows(data, timesteps, stride = 1):
    SET windows TO empty list
    FOR i FROM 0 TO (LENGTH of data - timesteps) STEP stride:
        SET window TO data SLICE from
            index i TO (i + timesteps)
        APPEND window TO windows
    END FOR

    RETURN windows
END FUNCTION
```

Where:

- `data`: represents the normalised time series data.
- `timesteps`: is the fixed length of each sliding window (250 in our case).
- `stride`: defines the step size for moving the window across the data.
- `window`: one singular window of fixed length.
- `windows`: list of overlapping windows.

### 2.2.3 TRAIN-TEST SPLIT

We split the dataset with a $80 : 20$ ratio with the following distribution of indices:

| Set | Index Range | Proportion of Data |
|---|---|---|
| Training Set | $0, 1, 2, \ldots, i-1$ | 80% |
| Testing Set | $i, i+1, i+2, \ldots, \texttt{len}(X) - 1$ | 20% |

Table 1: Time-series split of the dataset into training and testing sets.

### 2.3 BASELINE MODELS

To establish a benchmark for model performance, several baseline models were implemented. This allows us to assess the quality of CPC embeddings and whether they have extracted relevant features.

#### 2.3.1 PERSISTENCE MODEL

The persistence model is a naive, recursive-like forecasting method that assumes the future value of the time series is the most recent observed value. This can be expressed as:

$$\hat{y}_{t+1} = y_t \tag{3}$$

#### 2.3.2 ZERO MODEL

The zero model predicts that all future values log returns will be zero. This serves as a test for mean-reverting behaviour of the asset in question where returns periodically return to zero. We define it as:

$$\hat{y}_t = 0 \tag{4}$$

#### 2.3.3 MEAN MODEL

The mean model predicts that all future values will be equal to the mean of the observed training data. The predicted value, $\hat{y}$, is given by:

$$\hat{y} = \frac{1}{n} \sum_{i=1}^{n} y_i \tag{5}$$

where $n$ represents the number of observations in the training set.

#### 2.3.4 LINEAR REGRESSION MODEL

A linear regression model was used with lagged time series data as an input and non-lagged time series data as the target variable. We attempt to determine as a baseline if there is a linear relationship between the price at time $t - 1$ and the price at time $t$. The regression equation is represented as:

$$\hat{y} = \beta_0 + \sum_{j=1}^{p} \beta_j X_j \tag{6}$$

where $\beta_0$ is the intercept, $\beta_j$ are the coefficients for each lagged feature $X_j$, and $p$ is the number of lagged features.

### 2.4 LSTM MODEL

#### 2.4.1 OVERVIEW

To explore whether a simple model with our embeddings as input can outperform neural networks, a basic LSTM model was implemented. The LSTM model, first introduced by Sepp Hochreiter et al. (9) is a type of recurrent neural network (RNN) designed to be particularly suitable for sequential data. LSTMs are different to standard RNNs as they are capable of capturing both short-term and long-term patterns, helping to mitigate the vanishing gradient problem (9).

Through gate regulation, LSTMs maintain and update a memory cell state over time. This allows for the network to capture dependencies that span across many time steps in sequential data.

#### 2.4.2 ARCHITECTURE

The architecture for the LSTM is shown in Figure 3.

**Training Procedure:** The LSTM model was trained using the Mean Squared Error (MSE) loss function with the Adam optimiser used to update the model weights.

Due to the relatively large model size compared to the number of data points, training was done for a total of **40** epochs. This is to prevent possible overfitting on the train set.

## 2.5 CONTRASTIVE PREDICTIVE CODING (CPC)

CPC is an unsupervised representation learning approach designed to "learn the representations that encode the underlying shared information between different parts of the (high-dimensional) signal" (4). Unlike traditional supervised learning methods, CPC generates embeddings that capture the temporal structure and dependencies inherent in time series data.

### 2.5.1 OVERVIEW OF CONTRASTIVE PREDICTIVE CODING

The CPC's architecture's objective is to maximise the mutual information between a context vector and future observations in a latent space. The model consists of an encoder network that transforms raw inputs into a lower-dimensional representation and a context network that aggregates these representations over time to predict future sequences (4).

The CPC architecture implemented in this study is tailored for financial time series data and utilises windows as data points. The flow of data through the CPC model can be described as follows:

- **Encoder Network:** denoted as $f_\theta$, maps each window of raw input data, $X_i = (x_{i,1}, x_{i,2}, \ldots, x_{i,T})$, where $i$ denotes the window index and $T$ is the window length, into a lower-dimensional latent representation $z_i$. This process is represented by the equation:

$$z_i = f_\theta(X_i) \tag{7}$$

  where $z_i \in R^d$ is the latent representation of window $i$, and $f_\theta$ is a function parameterised by $\theta$ (in our case a series of one-dimensional convolutional layers followed by a dense layer).

- **Context Network:** denoted as $g_\phi$, aggregates the sequence of window embeddings $\{z_1, z_2, \ldots, z_n\}$ to produce a single context vector $c$ defined as:

$$c = g_\phi(z_1, z_2, \ldots, z_n) \tag{8}$$

  where $c \in R^h$ and $g_\phi$ is a function parameterised by $\phi$, implemented using a Gated Recurrent Unit (GRU). The GRU only returns the final hidden state to output a single context vector that has aggregated information from all window embeddings.

- **Contrastive Objective:** is to maximise the similarity between the context vector $c$ and the true future latent representation $z_{n+k}$, while minimising the similarity with the negative sample. We do this by minimising the binary cross-entropy loss function where the predicted probability $\hat{y}$ for the positive sample being the correct future latent representation is computed as:

$$\hat{y} = \sigma\left(\text{mean}(c \cdot z_{n+k})\right) \tag{9}$$

  where $\sigma$ denotes the sigmoid function. The binary cross-entropy loss, $L$, is then computed as:

$$L = -\frac{1}{N} \sum_{i=1}^{N} \left(y_i \log(\hat{y}_i) + (1 - y_i) \log(1 - \hat{y}_i)\right) \tag{10}$$

  where $y_i$ represents the binary label (1 for positive samples, 0 for negative samples), and $\hat{y}_i$ represents the predicted probability obtained from the sigmoid activation.

A diagram depicting the general CPC architecture can be seen in Figure 4, which was adapted from (4).

### 2.5.2 DATA GENERATION FOR CPC

The generation of negative samples is arguably the most important aspect of the CPC architecture. If negative samples are too easy to differentiate between positive samples, the model will fail to create relevant and predictive embeddings. In contrast, if negative samples are too similar to the positive samples, the model will fail to learn and high-quality embeddings will not be generated. Therefore, a custom data generator was developed to create batches of data specifically for contrastive learning. This generator produces sequences of time windows, with each batch containing both positive and negative pairs, along with corresponding labels indicating whether a given pair is a true or false match.

**Positive Sample Generation:** For each index or window in the time series data, the positive sample represents the next consecutive window of the time series.

The following pseudo-code illustrates the process for generating positive samples:

```
FOR each position i IN data LENGTH - (timesteps * (n_windows + 1)):
    SET context_windows TO empty list
    FOR each w IN range(n_windows):
        SET window TO data SLICE from
            (i+(w*timesteps)) TO (i + (w + 1) * timesteps)
        APPEND window TO context_windows
    SET y_positive TO data SLICE from
        (i+(n_windows*timesteps)) TO (i+(n_windows+1)*timesteps)
END FOR
```

**Negative Sample Generation:** Conventional CPC methodologies typically involve selecting random samples from different sequences or distant parts of the same sequence. In our approach, negative samples are generated using a normal distribution of random noise based on the parameters of the input window. This is based on the Black-Scholes framework which models financial time series as a stochastic process (10). Generating negative samples in this way allows for the model to learn to differentiate from random noise, therefore creating high-quality embeddings.

To generate a negative sample, a random normal distribution is created:

$$y_{\text{negative}} \sim \mathcal{N}(\mu_{y_{\text{negative}}}, \sigma_{y_{\text{negative}}}) \tag{11}$$

where $\mu_{y_{\text{negative}}}$ and $\sigma_{y_{\text{negative}}}$ are the mean and standard deviation of the input window.

The pseudo-code for generating negative samples is as follows:

```
FOR each position i IN data LENGTH - (timesteps * (n_windows + 1)):
    SET context_windows TO empty list
    FOR each w IN range(n_windows):
        SET window TO data SLICE from
            (i + w * timesteps) TO (i + (w + 1) * timesteps)
        APPEND window TO context_windows
    SET y_negative_base TO data SLICE from
        (i + (n_windows - 1) * timesteps) TO (i + n_windows * timesteps)
    GENERATE y_negative USING Gaussian noise WITH
        mean(y_negative_base) AND std(y_negative_base)
END FOR
```

This negative sampling strategy introduces noise that mimics random characteristics of financial time series, forcing the CPC model to learn patterns that are more predictive than randomness.

In Figure 2 one can see an example of an input window and its corresponding positive/actual sample and negative sample.

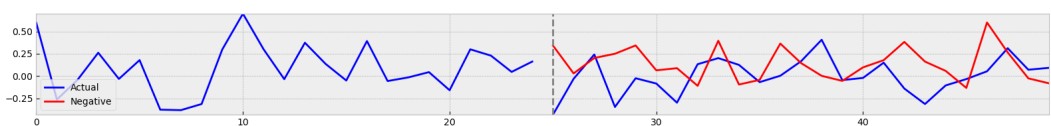

Figure 2: CPC Positive and Negative Sample Exampels

### 2.5.3 CPC MODEL ARCHITECTURE

The CPC model architecture is depicted in Figure 5. The architecture consists of an encoder network, which transforms input windows into embeddings, and a context network, which aggregates these embeddings over time to predict future observations.

### 2.5.4 TRAINING

During training, for every epoch the binary accuracy is displayed for both train and test set to monitor potential overfitting. The CPC architecture was ran for **100** epochs, after which overfitting to the train set was very apparent.

### 2.5.5 EVALUATION OF CPC EMBEDDINGS

After training, the encoder is frozen and used to create embeddings for both the training and testing datasets. These embeddings are then evaluated using the architectures mentioned earlier.

To analyse the embeddings from a visual standpoint, t-SNE, a dimensionality reduction technique was utilised to reduce the number of components to two. The K-means algorithm was then used to colour the embeddings in t-SNE vector space. Using the elbow method, it was determined that the ideal number of clusters was 4. Figure 6 shows the K-Means clustering.

Visually, one can clearly see distinct clustering and segregation of the different clusters. This suggests that the architecture is in fact learning and producing high-quality embeddings that effectively identify different groups in the input data.

### 2.6 SHARPE OPTIMISATION WITH LINEAR REGRESSION

To utilise the embeddings in a formal and financial context we have used our embeddings in combination with Linear Regression and evaluated its performance using the Sharpe Ratio.

### 2.6.1 SHARPE RATIO DEFINITION

William F. Sharpe defined the Sharpe ratio as: "the mean excess return per unit of standard deviation of excess return" (11). Mathematically, it is defined as:

$$\text{Sharpe Ratio} = \frac{E[R - R_f]}{\sigma_R} \tag{12}$$

where $R$ is the portfolio return, $R_f$ is the risk-free rate (often assumed to be zero for simplicity in some contexts), and $\sigma_R$ is the standard deviation of the portfolio returns. To analyse the Sharpe ratio, we multiply by $\sqrt{252}$, assuming 252 trading days in a year.

### 2.6.2 TRAINING & EVALUATION

We will evaluate our model on the unseen test set of the USDJPY dataset and compare the Sharpe ratio to a simple buy-and-hold strategy Sharpe ratio.

As a further test to show the quality of the embeddings generated, the encoder trained on USDJPY will then be used to generate embeddings on two other datasets: USDSGD and the EURGBP, to show our CPC architecture is capable of generating both generalist and specific features.

# 3 RESULTS

## 3.1 PERFORMANCE COMPARISON

The percentage difference is calculated using the formula:

$$\text{Percentage Difference} = \left( \frac{\text{Error of Model} - \text{Error of CPC-LR}}{\text{Error of CPC-LR}} \right) \times 100 \qquad (13)$$

## 3.2 LSTM MODEL PERFORMANCE

Despite the consensus that LSTMs are designed to avoid the gradient vanishing problem and should be able to capture temporal dependencies (9) , the results suggest that the LSTM model struggles to learn meaningful features from the USDJPY time series data. This can be attributed to the following factors:

- Financial time series are inherently noisy and are highly stochastic. This is due to a variety of factors (geopolitical events, macroeconomic indicators, etc.) that together contribute to an almost random behaviour. LSTM models require a substantial amount of data or features to learn effectively and therefore struggled with our task.

- LSTM architectures have more parameters than their non-neural network counterparts. If the number of parameters is significantly larger than the number of data points, overfitting has a high probability of occuring. This concern is exacerbated further when an LSTM is trained on noisy data.

## 3.3 LINEAR REGRESSION MODEL PERFORMANCE

The LR model, despite its simplicity and significantly fewer parameters compared to the LSTM model, outperforms the LSTM in forecasting the USDJPY. This surprising result can be explained by several reasons:

- Linear Regression, being a simpler model with fewer parameters, is less prone to overfitting compared to an LSTM. It does not attempt to capture complex non-linear patterns in the data, which can be beneficial due to the inherent noisy nature of the our input.

- The LR model provides stable parameter estimates even with relatively small datasets. This can be beneficial as the relationship between input and output might not be highly complex or might vary frequently due to external market conditions.

- The financial time series of exchange rates often exhibit linear trends or mean-reverting behaviour over short periods. The LR model, which inherently assumes a linear relationship between input variables and the output, can effectively capture these linear or near-linear trends without the need for complex model architectures.

## 3.4 MEAN AND ZERO MODELS: EVIDENCE OF MEAN-REVERTING BEHAVIOUR

Both the Mean and Zero models perform close to each other and have the lowest error rate among the other models. This outcome strongly suggests that the USDJPY rate exhibits a mean-reverting behaviour, a common characteristic in many financial time series.

- The Mean model, which predicts future values as the mean of the historical data, works well in scenarios where the time series is mean-reverting. The near-zero error indicates that the exchange rate often returns to its average return over time, making the Mean model a reliable predictor.

- Similarly, the Zero model, which assumes that all future log returns will be zero (implying no change in price), performs well, further supporting the hypothesis of mean reversion. A zero prediction effectively suggests that the current price level is expected to persist, which aligns with a market that lacks strong directional trends and frequently reverts to a mean.

## 3.5 SHARPE OPTIMISATION RESULTS

The table below presents the Sharpe ratios for different currency pairs using two different strategies: the LR model using CPC embeddings generated on the USDJPY dataset and the buy-and-hold strategy of the currency the embeddings are being evaluated on.

Table 2: Sharpe Ratios for Different Currency Pairs

| Currency Pair | Strategy Sharpe Ratio | Buy and Hold Sharpe Ratio |
|---|---|---|
| USDJPY | 1.312 | 0.5298 |
| USDSGD | 0.9802 | -0.3512 |
| EURGBP | 0.7405 | -0.1216 |

## 4 ANALYSIS

The strategy Sharpe ratios for all currency pairs evaluated are substantially higher than their corresponding buy-and-hold Sharpe ratios. This indicates that the CPC-based strategy is able to capture underlying patterns and signals in the data that are not apparent to traditional methods. The CPC model's ability to generalise and be fine-tuned on other currency pairs underscores its ability to learn general market sentiment.

In the plots below, allocations were scaled to be between -1 and 1. This does not affect the calculation of the Sharpe ratio and is done just for plotting purposes.

From Figure 8, the following is observed:

**Observations from Return Plots:** Upon examining the return plots, it is evident that the strategies exhibit an upward, almost linear trend on average. Additionally, there are sporadic jumps upwards in our strategies returns that coincide with significant movements in the original stock, as indicated by the returns of the benchmark buy-and-hold strategy. This pattern suggests that the features extracted by the model provide relevant information on which trades to execute and, more importantly, the optimal timing for these trades. The ability of the strategy to capture upward trends while also responding swiftly to substantial market movements indicates that the model effectively identifies profitable trading opportunities based on the underlying features.

**Analysis of Allocations:** The allocations plotted over time can be characterised by frequent fluctuations both upwards and downwards. This behaviour is consistent with the expected strategy of a market maker, whose role is to provide liquidity by constantly buying and selling, thereby ensuring a smooth market operation. The high-frequency nature of these allocations reflects an attempt to capture small price discrepancies, making quick trades that capitalise on these minor variations. Such behaviour is advantageous in forex markets, where tight spreads and high liquidity present numerous opportunities for rapid trades.

Furthermore, the allocations appear to be, on average, equally distributed between positive and negative positions indicating that the model is finding opportunities to go both long and short. This is a positive outcome as this type of neutral portfolio behaviour tends to perform well in the mean-reverting behaviour of foreign exchange markets. Since prices tend to return to a central value over time taking both long and short positions allows the model to profit from both upward and downward movements.

## 5 CONCLUSION

A CPC architecture was built that outputs relevant, high-quality embeddings for downstream financial forecasting time-series tasks. Negative samples in the architecture were generated non-traditionally using the Black-Scholes framework as a basis. These embeddings were used to successfully forecast several different foreign exchange currency pairs and beat all traditional benchmarks. Significant alpha was found demonstrated by a Sharpe ratio beating the traditional buy-and-hold

benchmark and ultimately helped in achieving our objective of producing an edge against the market.

In conclusion, the architecture developed can be used to generate features automatically without the need for manual feature engineering and domain expertise. This could impact further research into stochastic time series forecasting and improve the current state of the art.

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

# A  NEURAL NETWORK ARCHITECTURES

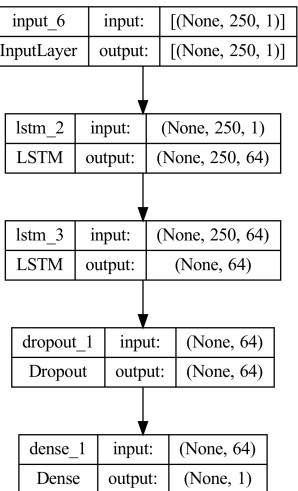

Figure 3: LSTM Architecture

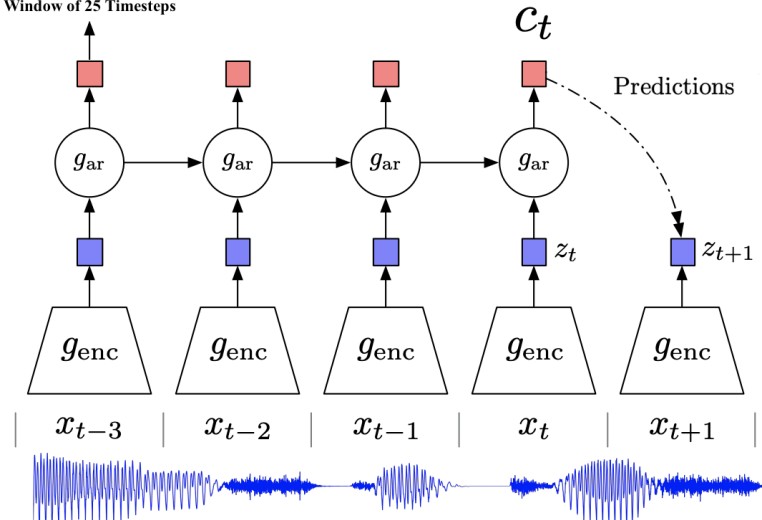

Figure 4: General CPC architecture diagram

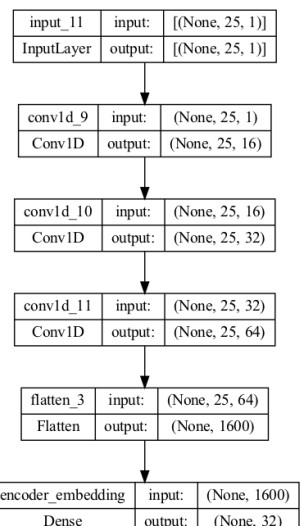

(a) Encoder Architecture

**Encoder:** The encoder model utilises three one-dimensional convolution layers with different sizes to extract relevant features from the input. This is then flattened to generate a single embedding for that window.

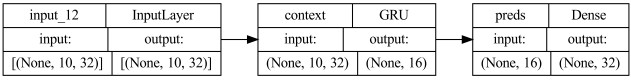

(b) Context Architecture

**Context:** The context model utilises one GRU layer to extract temporal features from the sequence of embeddings generated by the encoder from the windowed input.

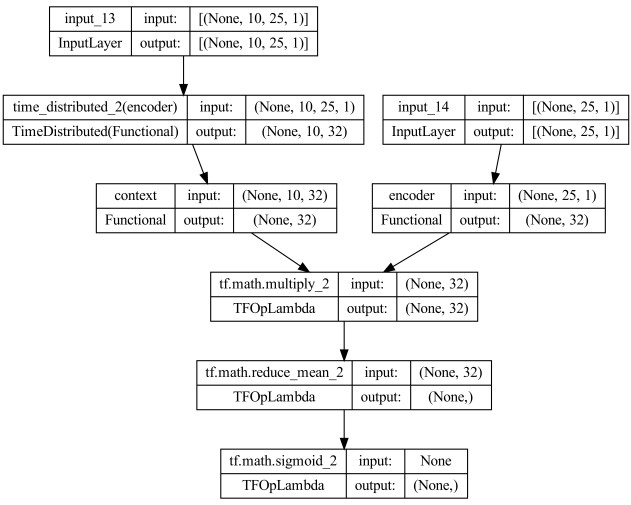

(c) Overall CPC Architecture

**Overall CPC Model:** The overall CPC model integrates the encoder and context models to predict future latent representations based on past input windows.

Figure 5: Overall CPC Model Architecture

## B  APPENDIX: THE LEARNING PROCEDURE

The following figures help visualise the progress of the learning procedure.

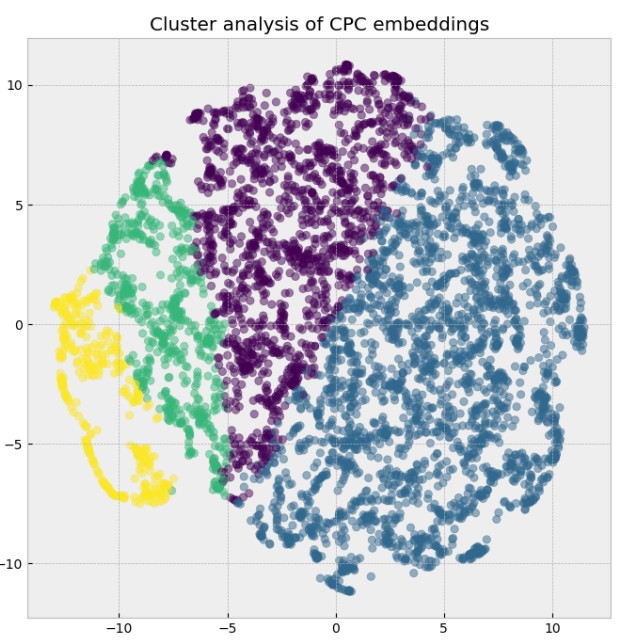

Figure 6: Clustering of embeddings in two-dimensions

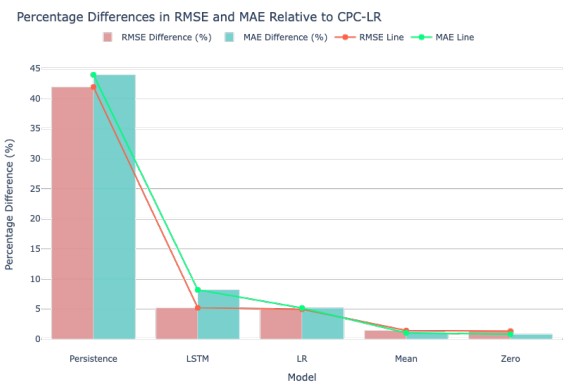

Figure 7: RMSE and MAE relative to CPC-LR

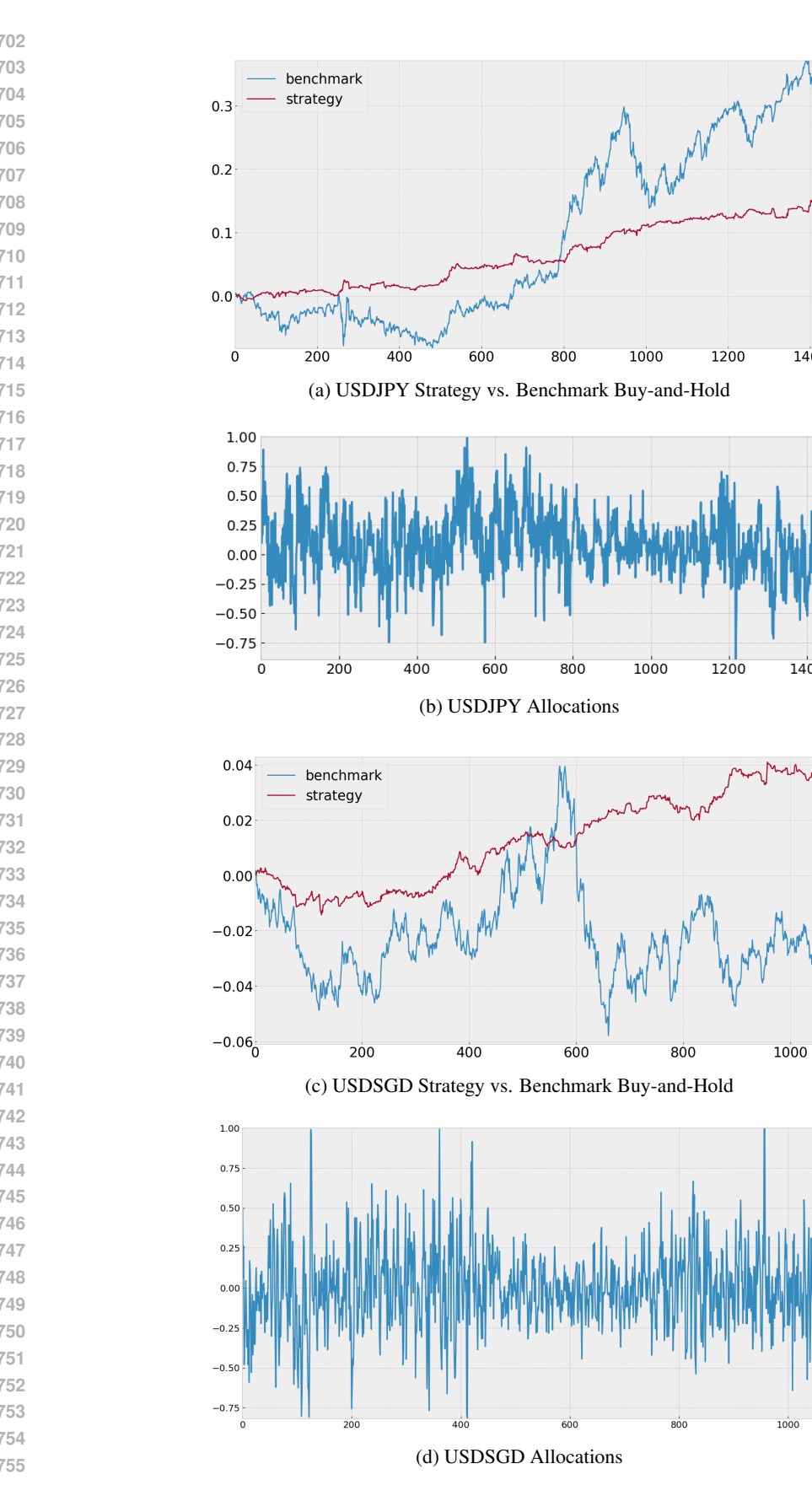

(a) USDJPY Strategy vs. Benchmark Buy-and-Hold

(b) USDJPY Allocations

(c) USDSGD Strategy vs. Benchmark Buy-and-Hold

(d) USDSGD Allocations

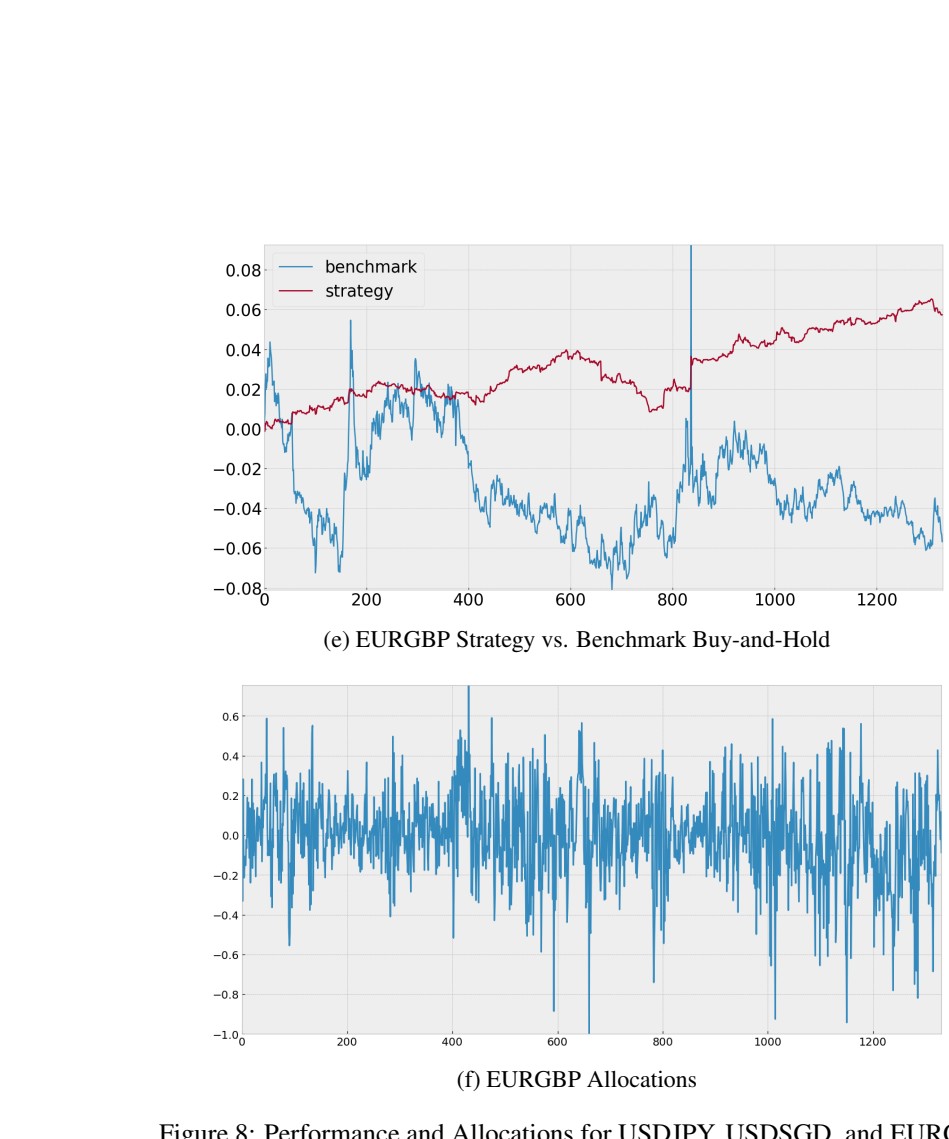

(e) EURGBP Strategy vs. Benchmark Buy-and-Hold

(f) EURGBP Allocations

Figure 8: Performance and Allocations for USDJPY, USDSGD, and EURGBP

