# OpenReview forum: "Representation learning for financial time series forecasting"
_ICLR.cc/2025/Conference — Submitted to ICLR 2025_

### Official Review · Reviewer_BdDD · 2024-10-25

**Soundness:** 1
**Presentation:** 1
**Contribution:** 2
**Rating:** 3
**Confidence:** 4

**Summary:**

The paper aims to overcome manual feature engineering in financial time series by focusing on automatic feature generation. To do so, it considers the application of unsupervised contrastive learning to the domain of financial time series. Specifically, the paper proposes an implementation of Contrastive Predictive Coding (CPC) to extract meaningful features from the data to be used in downstream forecasting tasks. Embeddings are optimized by maximizing the similarity with subsequent windows of observations (positive samples) while separating from random noise (negative samples). The extracted embeddings are paired with a linear regression model, improving performance over simple baselines in financial tasks.

Despite the interesting application and some promising results, I think the paper suffers from critical weaknesses that prevent me from recommending acceptance.

**Strengths:**

- Developing time series methods that are robust to inherent strong stochasticity in the data is significant.

- The application of CPC to the financial domain seems novel.

- Some of the presented results are promising.

**Weaknesses:**

Major:
- I did not find the paper’s motivations convincing. The first sections only mention that CPC was never evaluated in the financial domain. I think this is more of a necessary condition rather than sufficient. The paper would be much stronger if convincing arguments are provided on why CPC is a good choice for tackling stochasticity and why is preferable w.r.t. other unsupervised representation learning methods.

- The contextualization of this paper within the current literature is very poor. Only 3 references are provided in the literature review section: CPC, an application to anomaly detection, and an LSTM-based architecture. Overall, only 11 references are provided. I think the paper would benefit from a better presentation, up to recent works, of time series unsupervised representation learning [1] and forecasting [2], preferably discussing relevance to the financial domain.

- The presentation and organization of the paper’s content is very poor. A considerable portion of the paper (~3 pages in total) is dedicated to presenting naive baselines and details about the standard ML pipeline (preprocessing, normalization, windowing, train/test split), while all the important figures, describing the architecture and presenting the main results, are piled in the Appendix with minimal text. I think the authors should consider moving to the main text all tables/plots that they believe are relevant to support their claims.

-  The learned embeddings are only evaluated in conjunction with linear regression, which severely limits the impact of the work, especially given the targeted venue.

- The paper claims LSTMs struggling to learn from the considered dataset and being outperformed by linear regression and naive baselines (Sec. 3.2-3.3-3.4). However, Figure 7 (in Appendix) is the only result supporting this. Furthermore, the experimental pipeline does not consider a validation set for early stopping and model selection, which is arguably fundamental for deep architectures. Given the sensitivity of the paper's claim, I think it should be supported with more empirical evidence, e.g., learning curve, hyperparameter optimization.

Minor
- There are some inconsistencies in the notation. For example, $x_t$ is found on both sides of Eq.2;  notation is inconsistent between Eqs.1-2 and between Eqs. 3-4-5-6.

- The definition of the CPC-LR acronym is missing.

- Figure presentation could be improved, e.g., all plots in Fig. 8 are missing axis labels; captions are very concise; text in Fig.3-5-7 is small.

- The paper might benefit from a more gentle introduction to some domain-specific terminology, e.g., allocations, buy-and-hold, return plots.


[1] Meng, Qianwen, et al. "Unsupervised representation learning for time series: A review." arXiv preprint arXiv:2308.01578 (2023).

[2] Benidis, Konstantinos, et al. "Deep learning for time series forecasting: Tutorial and literature survey." ACM Computing Surveys 55.6 (2022).

**Questions:**

- What is preventing the authors from implementing the learned embeddings into neural architectures?
- Why do the authors think CPC is the right choice for tackling stochasticity in the data?
- Could the authors please clarify Figure 6? Leaving the coloring aside, I do not see any visual separation.

---

### Official Review · Reviewer_MF61 · 2024-10-26

**Soundness:** 1
**Presentation:** 1
**Contribution:** 1
**Rating:** 1
**Confidence:** 4

**Summary:**

This paper proposes to use contrastive predictive coding as a way of automating feature engineering for financial time series forecast. The authors integrated a contrastive representation learning technique with LSTM model and evaluated forecast on sharpe ratio metric.

**Strengths:**

The authors considered using contrastive representation learning to automate the feature engineering process.

**Weaknesses:**

- citation reference format is wrong
- there is no comparison with SOTA models or even vanilla transformers. there are only 11 references, authors should take a detailed look at the current landscape of literatures on financial time series forecasting
- the metric used are not standard time series forecasting metric such as MSE and MAE
- the experimental result is weak (only table 1)
- the presentation is poor, CPC is not explained in detail in related work. there are missing references in the paper. For example "In the plots below", there is no reference to which plot the authors are referring. there are many typos in the paper: normalise -> normalize, maximise -> maximize, and so on.
- there is a strong lack of novelty, CPC is already proposed and this work seems to be a simple application of CPC on financial time series.
- min-max data normalization may be problematic in financial time series because extreme values significantly affects the normalized data. Extreme values are highly prevalent in financial time series, but one extreme value could lead to 99% of the data compressed into a tiny range after min-max normalization.

**Questions:**

- the motivation is not clearly explained for using random noise as negative samples. The authors mentioned that negative samples should neither be too easy or too hard to distinguish from positive samples (ground truth samples). Then why is random gaussian noise considered suitable?

---

### Official Review · Reviewer_khLy · 2024-10-28

**Soundness:** 1
**Presentation:** 1
**Contribution:** 1
**Rating:** 1
**Confidence:** 5

**Summary:**

This work proposes to utilise an automated feature generation architecture, Contrastive Predictive Coding (CPC), to generate embeddings as input to improve the performance of downstream financial time series forecasting models.

**Strengths:**

The study tackles a pertinent research issue with possible academic or practical implications, serving as a solid foundation for further investigation. The endeavor to implement automated feature generation architecture to generate embeddings as input to improve the performance of downstream financial time series forecasting models. demonstrates an engagement with the subject, albeit certain aspects want improvement.

**Weaknesses:**

This paper has significant flaws in its writing, such as the absence of a proper literature review and problem formulation. Structurally, it resembles a technical report rather than a rigorous academic paper, lacking detailed descriptions of the methodology, adequate experiments to validate the model's effectiveness, and comprehensive information about the data used.

**Questions:**

1, The literature review is inadequate, with only 11 sources included, and many recent studies are missing. Additionally, the authors fail to identify and bridge the gap between the latest research and the proposed work. A thorough review of at least 40 relevant publications from the past five years is recommended, with a clear discussion of the research gaps and how this work contributes to the field.

2, All plots and tables are placed in the appendix with insufficient explanation and description. The paper should be restructured to follow a professional academic style, ensuring that visual elements are integrated within the main text and accompanied by detailed analysis.
The citation format is incorrect and needs to be revised. Additionally, there is missing information in citation 11, which should be completed to ensure consistency and accuracy.

3, It is unclear why the authors assume that using only the closing price as a feature is sufficient for model development. Stock prices are highly sensitive to external factors such as news, events, and economic indices. Relying solely on the closing price may lead to underfitting. If this risk exists, the authors should explain how they have addressed it.

4,The validation process relies exclusively on the Sharpe Ratio to assess the model’s performance. However, for a probabilistic forecasting model, additional evaluation metrics are required, such as point-wise measures (e.g., RMSE, MAE) and distributional similarity metrics (e.g., Wasserstein Distance). Please clarify the rationale for focusing solely on the Sharpe Ratio and consider including these additional metrics in the revised version.

5,The forecasting horizon is not clearly defined. Please specify whether this is a single-day-ahead forecast or a multi-step forecast, and provide further details.

6,The paper lacks necessary comparisons with state-of-the-art (SOTA) models. Please include at least four SOTA models from recent reputable journals or conferences, such as ICLR, the International Journal of Forecasting, and Applied Soft Computing, to provide context for the proposed work.

7, The paper lacks an ablation study, which is essential for understanding the contribution of each component of the model. Please add an ablation analysis in the revised version.

8. In the section conclusion, as author indicated the architecture developed can be used to generate features automatically without the
need for manual feature engineering and domain expertise. However, there is no experiments to validate this conclusion. Please add several experiments to validate the competitiness of the generated features compared to the other domain features.

**Details Of Ethics Concerns:**

In literatures part, it may be directly writen by LLM according to the style and format change.  I have to flag it and this part request more careful check.

---

### Official Review · Reviewer_P268 · 2024-10-31

**Soundness:** 2
**Presentation:** 1
**Contribution:** 1
**Rating:** 1
**Confidence:** 4

**Summary:**

This paper explores the use of Contrastive Predictive Coding (CPC) to enhance financial time series forecasting by generating automated feature embeddings that capture underlying market patterns without manual intervention. Given the limitations of traditional models in handling the noisy, stochastic nature of financial data, the authors benchmark CPC embeddings against both complex (LSTM) and simpler models (linear regression, persistence) on currency pairs like USDJPY, USDSGD, and EURGBP. Results demonstrate that CPC-enhanced models, especially linear regression, outperform traditional approaches, indicating that CPC can produce robust, generalizable features suitable for diverse currency pairs. Additionally, the study finds that CPC-based strategies yield higher Sharpe ratios than a buy-and-hold approach, suggesting CPC’s potential in developing adaptive trading strategies across financial markets.

**Strengths:**

1. While CPC itself is not new, this paper attempts to apply it in a challenging, stochastic domain—financial time series—which traditionally poses difficulties for predictive models due to noise and volatility. This application may contribute to the field by assessing CPC’s generalizability to non-deterministic data, even if further methodological enhancements are needed.
2. The paper evaluates CPC embeddings not just on a single currency but across multiple currency pairs (USDJPY, USDSGD, EURGBP), demonstrating an effort to test its adaptability. This cross-domain testing shows that the embeddings have the potential to capture broader, generalizable patterns, although more rigorous experimentation would solidify this claim.

**Weaknesses:**

1. The paper offers minimal novelty, as it relies on Contrastive Predictive Coding (CPC), a technique introduced in 2018, without significant modification. The sole methodological addition is the custom negative sampling based on the Black-Scholes model, which does not sufficiently advance. More substantial methodological innovation would be required for a contribution of significance.
2. The experimental scope is limited and omits comparisons with current state-of-the-art models for time series forecasting, such as iTransformer, PatchTST, and DLinear. Additionally, the paper does not benchmark CPC against recent self-supervised learning models tailored for time series data, like TimeDRL, TS2Vec, or TS-TCC. This lack of comparative analysis weakens the results’ credibility and makes it difficult to assess the true efficacy of the proposed approach.
3. The paper suffers from organizational and formatting issues, which detract from clarity and professionalism. Notably, the references do not adhere to the ICLR format, and sections are structured in an unconventional manner. For instance, the Related Work section should be distinct, and the Methodology section includes experimental details that would be better placed in a separate Experiments section. Improved structure and adherence to standard formatting would enhance the paper's readability and alignment with conference expectations.

**Questions:**

1. How does the CPC model in this paper address the transition from deterministic to stochastic environments, given the unpredictability of financial time series? Although the data is represented as stochastic via Black-Scholes-inspired negative sampling, the model lacks probabilistic forecasting, which could better capture uncertainty. Many models support probabilistic forecasting, such as estimating parametric distributions or quantiles—have the authors considered approaches like Bayesian neural networks or ensemble methods to quantify prediction uncertainty within the CPC framework?

---

### Official Review · Reviewer_UNnj · 2024-11-01

**Soundness:** 1
**Presentation:** 2
**Contribution:** 1
**Rating:** 3
**Confidence:** 4

**Summary:**

The work proposed a method for financial timeseries representation learning.

**Strengths:**

The description of model architecture is clear.

**Weaknesses:**

Literature
The work has limited literature research. I give some examples on related works:

1. Deep timeseries architectures
[1]: Zhou, Haoyi, et al. "Informer: Beyond efficient transformer for long sequence time-series forecasting." Proceedings of the AAAI conference on artificial intelligence. Vol. 35. No. 12. 2021.
[2]: He, Yangdong, and Jiabao Zhao. "Temporal convolutional networks for anomaly detection in time series." Journal of Physics: Conference Series. Vol. 1213. No. 4. IOP Publishing, 2019.
[3]: Wu, Haixu, et al. "Timesnet: Temporal 2d-variation modeling for general time series analysis." arXiv preprint arXiv:2210.02186 (2022).
[4]: Goswami, Mononito, et al. "Moment: A family of open time-series foundation models." arXiv preprint arXiv:2402.03885 (2024).

2. Timeseries representation learning
[1]: Eldele, Emadeldeen, et al. "Time-series representation learning via temporal and contextual contrasting." arXiv preprint arXiv:2106.14112 (2021).
[2]: Yue, Zhihan, et al. "Ts2vec: Towards universal representation of time series." Proceedings of the AAAI Conference on Artificial Intelligence. Vol. 36. No. 8. 2022.
[3]: Nie, Yuqi, et al. "A time series is worth 64 words: Long-term forecasting with transformers." arXiv preprint arXiv:2211.14730 (2022).

It is important to compare the proposed methodology with recently proposed representation learning technique to prove the effectiveness.

Experiment
1. Baseline comparison against existing representation learning method: In order to claim that the proposed method is effective, the author should also consider comparing the proposed method with some existing methods for timeseries representation learning, see corresponding section in literature above.

2. Ablation: The author should conduct ablation study on the pipeline to investigate the critical components that influence the performance. Some examples can be 1) varying model architecture for contrastive learning 2) experiment with different methods to create negative & positive samples for the proposed contrasitive learning method

3. Evaluation metric:
Market friction assumption: Can author provide the market friction assumption used during backtesting (such as transaction cost, assumed deviation from position entry, etc.) In addition, could you clarify whether your calculation of return for sharpe involves transaction cost? I ask this because in this type of trading strategy, despite the win-ratio (% profitable trade) might be higher than chance due to certain gain in predictive power, transaction cost could be very high and potentially eat up all the profit. This should be taken care of because the baseline, buy-and-hold, completes in two transactions.

Some other financial metrics to consider: max drawdown, sortino ratio. This gives the reader another perspective of the strategy from risk perspective.

Another note is that it seems like a position can be short only. If this is true, It would be helpful to provide margin needed at anytime just to be practical.

4. Size of dataset: Since the test conducted was only 20% of three currency timeseries. It would be helpful to backtest on more currency timeseries or other asset timeseries. This is to make sure the arrived results are statistically significant.

Additional Comment
1. Figure 6: It is not convincing that this plot shows the effectiveness of produced embedding. Essentially the unsupervised classes are plotted. Then, it is expected that the boundary from k-mean should be clear. The author should conduct some analysis on each cluster to see if there are unique properties for each of them.

2. lowest error rate with zero and mean model: It would be helpful for the author to provide more insights here. For example, does this suggest the forecasting method are worse than trivial baseline? If justified using mean-reverting property, can you show us that a forecasting model with mean-reverting behavior-forecast work well?

**Questions:**

See weaknesses

---

### Meta-Review · Area_Chair_2rxb · 2024-12-19

**Metareview:**

This paper has been evaluated by 5 knowledgeable reviewers. They have unanimously agreed that it does not meet the requirements of acceptance for ICLR (including 3 scores of strong rejection, and 2 straight rejects). The authors did not provide a rebuttal in response to the reviews, neither did they engage in a discussion with the reviewers.

**Additional Comments On Reviewer Discussion:**

The authors did not provide a rebuttal.

---

### Decision · Program_Chairs · 2025-01-22

Reject